# Review of Cathepsin K Inhibitor Development and the Potential Role of Phytochemicals

**DOI:** 10.3390/molecules30010091

**Published:** 2024-12-29

**Authors:** Dong Oh Moon

**Affiliations:** Department of Biology Education, Daegu University, 201, Daegudae-ro, Gyeongsan-si 38453, Gyeongsangbuk-do, Republic of Korea; domoon@daegu.ac.kr; Tel./Fax: +82-53-852-6992

**Keywords:** Cathepsin K, phytochemical, osteoporosis, inhibitor

## Abstract

Cathepsin K plays a pivotal role in bone resorption and has emerged as a prominent therapeutic target for treating bone-related diseases such as osteoporosis. Despite significant advances in synthetic inhibitor development, none have achieved FDA approval due to safety and efficacy challenges. This review highlights the potential of phytochemicals as alternative inhibitors, emphasizing their natural origin, structural diversity, and minimal adverse effects. Key phytochemicals, including AC-5-1, Cycloaltilisin 6, Cycloaltilisin 7, Nicolaioidesin C, and Panduratin A, were examined for their inhibitory activities against cathepsin K. While these compounds exhibit varying IC_50_ values, their docking studies revealed significant interactions within Cathepsin K’s active site, particularly involving critical residues such as Cys25 and His162. However, challenges such as lower potency compared to synthetic inhibitors and limited in vivo studies underscore the need for structural optimization and comprehensive preclinical evaluations. This review discusses biological insights, current limitations, and future strategies for advancing phytochemical-based inhibitors toward clinical applications in managing Cathepsin K-associated diseases.

## 1. Introduction

Cathepsins play a pivotal role in various biological processes, primarily through their proteolytic functions. These enzymes are categorized based on their catalytic mechanisms: Cathepsins B, C, F, H, K, L, V, O, S, W, and X are cysteine proteases, Cathepsins D and E are aspartyl proteases, and Cathepsins A and G belong to the serine protease family [1,2]. The enzymatic activity of cathepsins is intricately linked to numerous metabolic processes and has been associated with the progression of several diseases, as well as immune system responses [3,4]. Consequently, cathepsins have gained significant attention as potential therapeutic targets for various pathological conditions. For instance, cathepsin activity has been implicated in the development and progression of cancer, cerebrovascular accidents, Alzheimer’s disease, arthritis, chronic obstructive pulmonary disease (COPD), and glaucoma [5,6,7,8,9].

Among the cathepsins, Cathepsin K has emerged as a prominent therapeutic target, particularly in diseases related to bone resorption. This is due to its central role in the degradation of Type I collagen, the most abundant protein in the bone matrix. Cathepsin K is highly expressed in osteoclasts, the specialized cells responsible for bone resorption, and is secreted into the bone resorption lacunae, where it degrades collagen and other bone matrix components under acidic conditions. Unlike other proteases, Cathepsin K has unique enzymatic properties that allow it to effectively cleave collagen fibrils, making it a key player in bone remodeling processes [10,11].

Bone resorption is a fundamental physiological process mediated by osteoclasts, which degrade bone tissue to enable bone remodeling and maintain calcium homeostasis [10,11]. This tightly regulated process requires the coordinated activity of osteoclasts and osteoblasts to balance bone resorption and formation. However, imbalances in this equilibrium can lead to various skeletal disorders. For instance, excessive bone resorption relative to bone formation is the primary pathological mechanism underlying osteoporosis, a condition characterized by reduced bone mass and increased fracture risk. Osteoporosis is often aggravated by estrogen deficiency, which leads to elevated expression of receptor activator of nuclear factor kappa-Β ligand (RANKL), a key regulator of osteoclast differentiation and activity [12,13]. Similarly, rheumatoid arthritis and bone metastases involve localized bone destruction due to aberrant osteoclast activation.

This review focuses on osteoporosis, a major skeletal disorder driven by excessive bone resorption, and examines the physiological effects, clinical outcomes, and reasons for discontinuation of Cathepsin K inhibitors developed to date. In addition, the review explores research findings on phytochemical-based inhibitors of Cathepsin K, highlighting their potential as alternative therapeutic agents. However, it also critically evaluates the challenges associated with their development and discusses strategies to overcome these limitations.

Before delving into the inhibitors, it is crucial to understand the expression patterns and structural characteristics of Cathepsin K, which form the foundation for its therapeutic targeting.

## 2. Bone Resorption and Cathepsin K Expression

Bone resorption is a process driven by osteoclasts, which are large, multinucleated cells originating from hematopoietic stem cells within the monocyte/macrophage lineage. Once activated, osteoclasts attach to the bone surface, creating a specialized microenvironment called the resorption lacuna. In this sealed compartment, osteoclasts actively pump hydrogen ions (H^+^) through H^+^-ATPase and transport chloride ions (Cl^−^) via chloride channels, establishing an acidic milieu necessary for breaking down the inorganic component of hydroxyapatite. Simultaneously, lysosomal enzymes, notably Cathepsin K, are secreted to degrade the organic matrix, which consists predominantly of Type I collagen [14].

The degradation products of bone matrix breakdown, including calcium ions (Ca^2+^), phosphate ions (PO_4_^3−^), and collagen fragments, are internalized by osteoclasts through endocytosis or membrane transport mechanisms. These degradation products are then processed within the osteoclasts and subsequently released into the extracellular space, where they can be reabsorbed by neighboring cells or transported via the bloodstream to maintain systemic mineral homeostasis and support other physiological processes.

The human Cathepsin K gene spans approximately 12.1 kilobases of genomic DNA and is located on chromosome 1q21 [15]. Analysis of its genomic structure reveals that the gene comprises eight exons and seven introns. The resulting transcriptional product is 1.7 kilobases in length. Notably, the promoter region lacks the typical TATA or CAAT boxes near the transcriptional initiation site. Instead, it contains two consensus Sp1-binding sites and a G-rich region (42.5%) that may function as regulatory elements. Primer extension analysis has identified the transcription start site to be 58 base pairs upstream of the methionine start codon. The initiation of transcription appears to be regulated by several predicted transcription factors, including AP1, AP3, H-APF-1, PU.1, ETS-1, PEA3, MITF, NF-κB, and TFE3 [16,17,18].

Cathepsin K, a cysteine protease belonging to the papain-like family, is highly expressed in osteoclasts and is essential for breaking down the bone matrix. Osteoblasts and stromal cells release key cytokines such as the macrophage-colony-stimulating factor (M-CSF) and RANKL to drive osteoclast differentiation [19,20]. RANKL interacts with its receptor RANK on osteoclast precursors, activating intracellular-signaling pathways through tumor necrosis factor receptor-associated factors (TRAFs) [21]. This interaction stimulates several downstream-signaling mechanisms, including the MAPK, NF-κB, Src, and Akt pathways, which are critical for osteoclast formation and function [22,23,24,25].

Notably, RANKL upregulates the expression and activity of Cathepsin K in a dose- and time-dependent manner [26,27]. Additionally, numerous factors that regulate RANKL secretion from osteoblasts and stromal cells also impact the expression of Cathepsin K. These stimulatory influences include Vitamin D, parathyroid hormone, TNF-α, glucocorticoids, IL-1, IL-11, thyroid hormone, Prostaglandin E2, lipopolysaccharide, fibroblast growth factor-2, histamine, insulin-like growth factor-1, and even microgravity conditions [28,29]. In contrast, RANKL production is inhibited by estrogen and transforming growth factor-β (TGF-β), underscoring the complex regulatory network involved [30,31].

RANKL regulates Cathepsin K gene transcription through several mechanisms. An early event in this process involves the activation of TRAF6, a key adaptor protein in RANK signaling. Overexpression of TRAF6 enhances Cathepsin K promoter activity, while dominant negative forms of TRAF6 suppress this activation [18]. Additionally, the transcriptional activation of Cathepsin K is modulated by AP-1, a transcription factor complex formed by c-Fos and c-Jun. JunB alone can enhance basal Cathepsin K promoter activity, whereas c-Jun, JunD, and c-Fos require co-transfection with other Jun family members to significantly amplify promoter activity [17]. Silencing of c-Jun or JunB via siRNA suppresses RANKL-mediated Cathepsin K expression, highlighting the critical role of AP-1 in this regulatory process.

Further downstream, RANKL induces the phosphorylation of NFAT2 via p38 MAPK [32]. Phosphorylated NFAT2 translocates to the nucleus, where it activates Cathepsin K transcription. This mechanism differs from the classical calcineurin-mediated dephosphorylation of NFAT proteins, suggesting a unique regulatory paradigm for NFAT2 in osteoclasts. Additionally, RANKL activation leads to the phosphorylation of microphthalmia-associated transcription factor (MITF) through the p38-signaling pathway. Phosphorylated MITF binds to E-box sequences within the Cathepsin K promoter region, and its overexpression has been shown to markedly increase Cathepsin K expression in cultured osteoclasts [33,34].

Other factors, such as retinoic acid, periodic mechanical stretching, and extracellular matrix proteins (e.g., collagen Type I, fibronectin, vitronectin, and osteopontin), also promote Cathepsin K expression [29,35]. Conversely, physiological inhibitors like osteoprotegerin (OPG), interleukin-6 (IL-6), and interferon-γ (IFN-γ) can suppress Cathepsin K expression, reflecting the dynamic regulation of bone resorption [36]. The processes of bone resorption and the regulation of Cathepsin K expression are illustrated in Figure 1.

Expressed cathepsins are not restricted to specific cellular organelles. Instead, they move between phagosomes, endosomes, and lysosomes, playing diverse roles [37,38]. Under certain physiological conditions, some cathepsins may accumulate within specific organelles. Additionally, lysosomal membrane damage caused by external oxidants, such as reactive oxygen species (ROS), can lead to the release of cathepsins into the cytoplasm [39].

Cathepsin K mRNA is expressed across a range of tissues, including the bone, ovary, heart, placenta, lung, skeletal muscle, colon, and small intestine [7,40]. Its expression is particularly elevated in osteoclasts and osteoclast-like multinucleated giant cells [41,42]. Research has confirmed that Cathepsin K is essential for facilitating bone resorption, highlighting its critical function in skeletal maintenance and remodeling processes.

## 3. Structure and Mechanism of Cathepsin K

Cathepsin K is a lysosomal cysteine protease that plays a critical role in the degradation of Type I collagen, the primary component of the bone matrix [40]. As a member of the papain-like cysteine protease family, Cathepsin K is predominantly expressed in osteoclasts, where it facilitates bone resorption. Its unique ability to cleave collagen triple helices at multiple sites distinguishes it from other cysteine proteases and underscores its importance in maintaining bone homeostasis.

Cathepsin K is synthesized as an inactive pre-proenzyme consisting of 329 amino acids, with a molecular weight of approximately 38 kDa [43,44]. This precursor protein comprises three distinct regions: a 15-amino acid signal sequence, a 99-amino acid propeptide, and the 215-amino acid mature catalytic domain. The signal sequence directs the nascent protein to the rough endoplasmic reticulum (RER) and is cleaved post-translation within the RER to facilitate proper localization. The propeptide region of Cathepsin K contains a conserved N-glycosylation site, which is essential for targeting the inactive enzyme to lysosomes via the Golgi apparatus. This targeting occurs through the mannose 6-phosphate receptor pathway, a universal mechanism for lysosomal enzyme transport. During activation within the lysosome, the 99-amino acid propeptide is cleaved at the specific site between Arg114 and Ala115, producing the mature enzyme composed of 215 amino acids [45]. This maturation process is crucial for enabling the enzyme’s full catalytic activity, which is necessary for functions such as collagen degradation. The biosynthesis, processing, and active site structure of Cathepsin K are shown in Figure 2.

The catalytic domain of Cathepsin K is organized into two distinct folded domains that together form a unique “V”-shaped cleft. This cleft constitutes the enzyme’s active site, where substrate binding and catalysis occur. The left domain is distinguished by a central helix as its primary structural feature, while the right domain is dominated by beta-barrel motifs. The active site is strategically located at the interface of these two domains, facilitating precise interactions with substrates and enabling efficient enzymatic activity.

The mature form adopts a papain-like fold, and the active site of Cathepsin K contains a catalytic dyad comprising Cys25 and His162 [46]. These residues form a V-shaped cleft that facilitates its collagenolytic activity. In addition, residues Tyr67 and Leu208 play crucial roles in defining the distinctive substrate specificity of Cathepsin K. These structural features enable Cathepsin K to effectively interact with and cleave collagen substrates.

The catalytic mechanism of Cathepsin K involves a thiolate–imidazolium ion pair, where the cysteine residue at Position 25 (Cys25) acts as a nucleophile. This nucleophilic attack on the carbonyl carbon of the peptide bond results in the formation of a tetrahedral intermediate. The intermediate is stabilized by the oxyanion hole, facilitating peptide bond cleavage and subsequent hydrolysis [47]. This mechanism enables Cathepsin K to degrade collagen by cleaving within the triple helix. Cathepsin K interacts with collagen at Gly–Pro–Xaa sequences, where Xaa is often proline or hydroxyproline, allowing cleavage at multiple internal sites of Type I collagen [48,49]. The catalytic mechanism of Cathepsin K in collagen degradation is shown in Figure 3.

Cathepsin K is also subject to oxidative regulation, where reactive oxygen species can oxidize the catalytic cysteine, rendering the enzyme inactive. Its positively charged surface enhances interactions with glycosaminoglycans (GAGs), such as chondroitin 4-sulfate, which provide partial protection against oxidative damage [50]. Cathepsin K operates optimally in an acidic environment, with its activity peaking at pH levels typical of lysosomes and the osteoclast resorption lacuna. This acidic microenvironment, generated by the activity of H^+^-ATPase, is essential for both the solubilization of hydroxyapatite and the activation of Cathepsin K [51].

By combining structural adaptability with environmental sensitivity, Cathepsin K is uniquely equipped to facilitate the efficient breakdown of collagen in bone resorption, a critical process for maintaining skeletal health and remodeling.

## 4. Cathepsin K Inhibitors: Key Characteristics, Challenges, and Future Directions

Cathepsin K inhibitors have gained significant attention as potential therapeutic agents for the treatment of osteoporosis and other bone-related disorders due to their ability to selectively target the collagen-degrading activity of osteoclasts. Various Cathepsin K inhibitors have been developed and advanced into clinical trials. However, their comprehensive characteristics and study outcomes have already been summarized in earlier reviews [52,53,54]. Therefore, this review focuses on two aspects: the key characteristics of Cathepsin K inhibitors under development and the summary of their preclinical and clinical results, as illustrated in Table 1 and Table 2, respectively. Table 1 highlights critical molecular and pharmacological attributes of prominent inhibitors, such as odanacatib, balicatib, and ONO-5334, while Table 2 outlines the findings from studies conducted across various stages of drug development, including preclinical evaluations and Phase I, II, and III trials.

Despite promising efficacy in improving bone mineral density (BMD) and reducing bone resorption markers, the development of Cathepsin K inhibitors has faced multiple setbacks. Most notably, no Cathepsin K inhibitor has successfully achieved FDA approval to date [55]. Odanacatib, developed by Merck, demonstrated substantial efficacy in clinical trials by increasing BMD and reducing fracture risks [56,57]. However, it was discontinued due to safety concerns, including a slight but significant increase in stroke risk [58]. Similarly, balicatib, developed by Novartis, showed potent inhibition of Cathepsin K and improvements in BMD but was terminated during development due to off-target effects, such as lysosomotropic accumulation leading to morphea-like skin thickening [59]. ONO-5334, another promising inhibitor, displayed favorable preclinical and clinical results, but its current development status remains unclear, with no updates on ongoing trials. The inability of these inhibitors to balance efficacy and safety has been a major obstacle to their advancement.

The challenges faced by Cathepsin K inhibitors underline the need for innovative approaches to ensure their clinical success. First, enhanced selectivity for Cathepsin K over other cathepsins must be achieved to minimize off-target effects, particularly in tissues such as skin and immune cells. Advanced computational modeling and structure-based drug design can aid in optimizing inhibitors for higher specificity. Second, strategies to mitigate adverse effects, such as the lysosomotropic properties observed with balicatib, should be integrated into early drug design processes. Developing compounds with non-basic scaffolds or incorporating prodrug strategies may address this issue. Third, the potential for combination therapies should be explored. Combining Cathepsin K inhibitors with agents that target other pathways involved in bone remodeling, such as RANKL inhibitors or anabolic treatments like parathyroid hormone analogs, could enhance therapeutic outcomes while reducing adverse events. Finally, a deeper understanding of the biological mechanisms linking Cathepsin K activity to systemic effects, such as cardiovascular risks, is essential. This knowledge can inform patient stratification strategies to identify populations most likely to benefit from these inhibitors while minimizing harm. As the field progresses, leveraging multidisciplinary approaches that integrate medicinal chemistry, pharmacology, and clinical insights will be key to overcoming the challenges and realizing the therapeutic potential of Cathepsin K inhibitors.

**Table 1 molecules-30-00091-t001:** Key characteristics of Cathepsin K inhibitors under development. “N.T.” stands for “not tested”.

Inhibitor	Developer	Year Developed	Ki (nM)	IC_50_ (nM)	Ref.
Relacatib 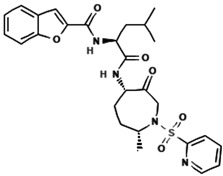	GlaxoSmithKline (Brentford, UK)	2006	Cathepsin K: 0.041 Cathepsin L: 0.068 Cathepsin V: 0.053	Cathepsin K (in situ): 45 Osteoclastic bone resorption inhibition: 70	[60,61]
Balicatib 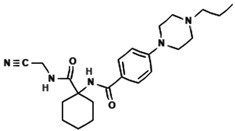	Novartis (Basel, Switzerland)	2005	Cathepsin K: 0.8 Cathepsin L: 5.5 Cathepsin B: 6.2	Cathepsin K: 1.4 Cathepsin L: 48 Cathepsin B: 56 Cathepsin S: 2900	[62]
Odanacatib 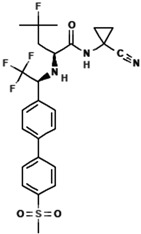	Merck (Darmstadt, Germany)	2004	Cathepsin K: 0.2	Cathepsin K: 0.2 Cathepsin B: 1034 Cathepsin L: 2995 Cathepsin S: 60	[63]
ONO-5334 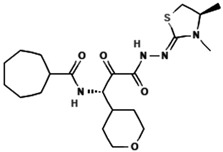	Ono Pharmaceutical (Osaka, Japan)	2011	Cathepsin K: 0.10 Cathepsin S: 0.83 Cathepsin L: 1.7 Cathepsin B: 32	N.T	[64]

**Table 2 molecules-30-00091-t002:** Summary of Cathepsin K inhibitor studies across preclinical and clinical phases.

Inhibitor	Phase	Findings	Ref.
L-235	Preclinical	- Prevented bone mineral density (BMD) loss at the lumbar spine in ovariectomized rabbits. - Did not reduce trabecular mineralizing surfaces or endocortical formation rates, preserving normal bone formation.	[65]
Relacatib	Preclinical	- Reduced bone resorption and formation markers in ovariectomized monkeys - Preserved osteonal and periosteal bone formation at cortical sites.	[61]
	Phase I	- Evaluated for effects on metabolism of ibuprofen, acetaminophen, and atorvastatin. - Development stopped after Phase I due to concerns over drug–drug interactions.	[61]
Balicatib	Preclinical	- Partially prevented bone loss, reduced bone turnover, and increased periosteal bone formation in ovariectomized monkeys.	[66]
	Phase II	- Increased tartrate-resistant acid phosphatase (TRAcP) accumulation in osteoclasts due to inhibited extracellular collagen degradation, leading to unique morphological changes in osteoclasts (“grape-like” appearance). - Development was halted due to lysosomal accumulation causing adverse skin reactions.	[67]
ONO-5334	Preclinical	- Improved cortical BMD and bone mineral content (BMC) in ovariectomized rodents and monkeys. - Dose-dependent reductions in bone resorption markers. - Less effective than alendronate for trabecular BMD but superior for cortical BMD.	[68]
	Phase I	- Plasma ONO-5334 reached a steady state within 2 days. - At 15 days, 100 mg, 300 mg, and 600 mg daily reduced urinary C-terminal telopeptide of Type I collagen (CTX) by 44.9%, 84.5%, and 92.5%, respectively. - Minimal effects on bone-specific alkaline phosphatase (B-ALP), osteocalcin (OC), and TRAP5b levels, even after 28 days.	[69]
	Phase II	- Significantly increased BMD at lumbar spine, total hip, and femoral neck after 24 months. - Bone formation markers (B-ALP, PINP) returned to near-baseline levels after initial suppression. - No clinically relevant safety concerns were observed.	[70]
Odanacatib	Preclinical	- Reduced osteoclast numbers and bone loss in periodontitis lesion areas.	[71]
	Phase I	- Demonstrated a long half life (66–93 h), enabling weekly dosing. - Dose-dependent reductions in CTX and N-telopeptides of cross-links of Type I collagen (NTX) markers of bone resorption. - Well tolerated with no significant adverse events reported.	[72]
	Phase II	- Dose-dependent increases in BMD at the lumbar spine and femoral neck (up to 5.7% at the spine). - Bone resorption markers declined rapidly; formation markers showed delayed changes. - Safe with minor adverse events like uncomplicated urinary infections.	[73,74]
	Phase III	- LOFT trial demonstrated significant fracture risk reduction (54% vertebral, 47% hip). - BMD increases sustained over years (+10.9% lumbar spine). - Development stopped due to increased risks of cerebrovascular accidents and incidence of stroke.	[75,76]

## 5. Phytochemicals to Inhibit Cathepsin K Activity

The search for effective Cathepsin K inhibitors has increasingly focused on phytochemicals due to their natural origin, structural diversity, and lower risk of adverse effects compared to synthetic compounds. Phytochemicals, defined as bioactive compounds derived from plants, have shown remarkable promise in drug discovery because of their unique chemical frameworks, which often allow for interactions with diverse biological targets [77,78]. Notable examples of phytochemical-based drugs include paclitaxel (Taxol^®^) from *Taxus brevifolia*, used in cancer therapy, and artemisinin from *Artemisia annua*, a cornerstone for malaria treatment [79,80]. These examples illustrate the potential of phytochemicals to inspire new therapeutic approaches across a range of diseases.

Phytochemicals provide an untapped resource for developing therapeutics to address Cathepsin K-mediated diseases such as osteoporosis, arthritis, and other conditions involving pathological bone resorption. However, despite their potential, phytochemicals often face limitations such as relatively lower potency compared to synthetic inhibitors. Nonetheless, their favorable safety profiles and structural diversity make them attractive candidates for further development. Below, several phytochemicals with inhibitory effects on Cathepsin K are summarized, highlighting their source, chemical structure, and biological activity.

### 5.1. AC-5-1

AC-5-1 is a dihydrochalcone derivative isolated from the bud covers of *Artocarpus altilis* (breadfruit) during a bioassay-guided study on natural products targeting Cathepsin K. This compound possesses a molecular formula of C_25_H_30_O_5_ and a molecular weight of 410 Da. Structural elucidation using NMR and mass spectrometry revealed the presence of a geranyl side chain and a dihydrochalcone core. With an IC_50_ value of 170 nM, AC-5-1 exhibits strong inhibitory activity against Cathepsin K, making it a promising candidate for further investigation [81]. Further analysis of AC-5-1 revealed that the compound contains a hydrogen-bonded phenolic proton at δ 12.81 and displays an aromatic ABX spin system in its NMR spectrum. The COSY and heteronuclear correlations confirmed the presence of seven methines, five methylenes, and three methyl groups. The dihydrochalcone moiety with a ketone carbonyl at δ 204.0, along with a geranyl side chain, were key structural elements elucidated through 13C NMR data.

### 5.2. Cycloaltilisin 6

Cycloaltilisin 6, another compound derived from *Artocarpus altilis*, is a dimeric dihydrochalcone that demonstrates potent Cathepsin K inhibitory activity with an IC_50_ value of 98 nM. Its structure consists of two AC-5-1 monomeric units asymmetrically linked to the C-12 and C-11′ positions of their B-rings. The molecular formula of Cycloaltilisin 6 is C_50_H_58_O_10_, with a molecular mass of 818 Da. Advanced NMR techniques, including COSY, HMQC, and HMBC, were used to confirm its dimeric structure. This structural dimerization significantly enhances its bioactivity, underscoring its potential as a lead compound for further development [81]. Detailed structural analyses revealed two hydrogen-bonded phenolic protons at δ 12.81 and δ 12.75 in the NMR spectrum of Cycloaltilisin 6. The presence of two aromatic ABX spin systems and two aromatic singlets further supports its asymmetric dimeric structure. Additionally, HMBC correlations confirmed the linkage between C-12 and C-11′ of the two AC-5-1 monomeric units. This asymmetric bridging increases the number of unique NMR signals, resulting in a total of 50 carbon resonances, which includes two ketone carbonyl signals at δ 205.7 and δ 204.9.

### 5.3. Cycloaltilisin 7

Cycloaltilisin 7, a prenylated flavone derivative isolated from *Artocarpus altilis*, has a molecular formula of C_25_H_26_O_5_ and a molecular weight of 406 Da. Structural elucidation revealed the presence of a flavone skeleton with an additional pyran ring formed by the cyclization of a prenyl group at the C-7 position. This unique structure distinguishes Cycloaltilisin 7 from other compounds in the series. The compound demonstrates moderate inhibitory activity against Cathepsin K, with an IC_50_ value of 840 nM. The structural features of Cycloaltilisin 7 were confirmed through NMR and HMBC analyses. The NMR spectrum revealed olefinic proton signals at δ 5.53 and δ 6.54, which are indicative of a cis-coupled double bond within the pyran ring. Key HMBC correlations linked the pyran ring to the flavone nucleus, confirming the attachment of the cyclized prenyl group at the C-7 position. Additionally, the spectrum displayed characteristic signals corresponding to phenolic hydroxyl groups and other substituents on the flavone skeleton. Despite being less potent than Cycloaltilisin 6, the unique structural features of Cycloaltilisin 7, such as the pyran ring and prenyl substitution patterns, contribute to its moderate inhibitory activity. These findings highlight the potential of Cycloaltilisin 7 as a valuable lead compound for studying the structure–activity relationship in Cathepsin K inhibitors.

### 5.4. Nicolaioidesin C

Nicolaioidesin C, a cyclohexenyl chalcone derivative isolated from the rhizomes of *Boesenbergia pandurata* (fingerroot), naturally occurs as a racemic mixture. With an IC_50_ value of 4.4 µM against Cathepsin K, Nicolaioidesin C shows moderate inhibitory activity compared to synthetic inhibitors. Its chemical structure, featuring a cyclohexenyl chalcone core, is critical for its ability to modulate protease function. Additionally, it exhibits potent inhibitory effects on cathepsin L in a cell-free enzyme assay (IC_50_ = 1 µM). The structural features of Nicolaioidesin C make it a promising scaffold for future optimization [82,83].

### 5.5. Panduratin A

Panduratin A, also derived from *Boesenbergia pandurata*, is a cyclohexenyl chalcone that exhibits broad-spectrum inhibitory activity against multiple cysteine proteases, including Cathepsins G, H, K, L, S, V, and X. Specific studies have demonstrated IC_50_ values of 5.1 µM and 6.6 µM for the enantiomers (+)-panduratin A and (−)-panduratin A, respectively, in Cathepsin K assays. Importantly, it shows selective inhibition of cathepsins without affecting other protease classes, such as matrix metalloproteases or caspases, highlighting its specificity. These findings make Panduratin A an interesting lead compound for therapeutic development [83]. Panduratin A’s broad-spectrum inhibitory activity and specificity towards cysteine proteases highlight its potential for therapeutic development in protease-associated diseases.

### 5.6. HIF (Hinokiflavone)

HIF, a biflavonoid isolated from the leaves of *Cycas guizhouensis*, demonstrates inhibitory effects against Cathepsin K (IC_50_ = 1.54 µM) and Cathepsin B (IC_50_ = 0.58 µM). Its structure comprises two flavonoid units linked by an oxygen bridge, enhancing its ability to interact with cathepsin active sites. The dual inhibition of Cathepsins B and K suggests potential therapeutic applications in diseases involving protease dysregulation, such as osteoporosis and cancer [84]. Flexible docking studies confirmed that HIF interacts strongly with key residues in Cathepsin B, including His110, His111, and Cys29. These interactions are critical for its high affinity and specificity towards Cathepsin B’s endopeptidase activity. HIF acts as a reversible inhibitor, as confirmed by time-dependent kinetic studies that showed its inhibitory effect stabilizes over time. This dual inhibition of Cathepsins B and K suggests potential therapeutic applications in diseases involving protease dysregulation, such as osteoporosis and cancer.

### 5.7. AMF4

AMF4, another biflavonoid from *Cycas guizhouensis*, demonstrates reversible inhibitory activity against Cathepsins B and K, with an IC_50_ of 1.39 µM for Cathepsin K. Its structural flexibility, owing to saturation at the 2,3 positions, allows for improved binding within enzyme active sites. AMF4’s moderate potency highlights its potential as a scaffold for structural refinement to enhance its therapeutic efficacy [84]. Docking studies revealed that AMF4 interacts with both the active site and occluding loop residues of Cathepsin B, including His111, Gly121, and Cys119. These interactions enable AMF4 to inhibit both the endopeptidase and exopeptidase activities of Cathepsin B, with endopeptidase inhibition being significantly stronger. This dual activity and its moderate potency highlight AMF4’s potential as a scaffold for further development of selective inhibitors targeting cathepsins.

### 5.8. 6-Shogaol

Furthermore, 6-Shogaol, a phenolic compound derived from the rhizomes of *Zingiber officinale* (ginger), acts as an uncompetitive inhibitor of Cathepsin K with a Ki value of 16.6 µM. Formed through the dehydration of gingerol during drying processes, it exhibits diverse biological activities, including anti-inflammatory and anti-resorptive effects. While its potency is lower compared to synthetic inhibitors, its natural origin and safety profile make it an attractive candidate for further development [85,86]. Moreover, 6-Shogaol exhibits anti-catabolic activity by inhibiting matrix metalloproteinase (MMP)-2 and MMP-9 activities in chondrocytes, which are critical enzymes involved in cartilage degradation. In addition, 6-Shogaol’s Cathepsin K inhibition occurs through an allosteric mechanism, selectively targeting the enzyme–substrate complex without affecting the free enzyme. This inhibitory profile distinguishes 6-Shogaol from other natural inhibitors of Cathepsin K, emphasizing its potential as a therapeutic agent for cartilage and bone-related diseases.

### 5.9. Challenges and Future Directions

The references in this section primarily date back from the early 2000s to the early 2010s, a period marked by significant efforts to explore natural products as inhibitors of Cathepsin K. Foundational studies, such as those by Patil et al. (2002) and Zeng et al. (2006), laid the groundwork by elucidating the structural and functional interactions between phytochemicals and Cathepsin K [81,84]. However, over the past decade, interest in this area has waned, with research priorities shifting toward synthetic inhibitors and advanced drug discovery platforms.

Despite this shift, the earlier studies remain highly relevant, as they provide valuable insights into the bioactivity and structural diversity of phytochemicals. Compounds such as dimeric dihydrochalcones, biflavonoids, and cyclohexenyl chalcones have demonstrated the potential to modulate Cathepsin K activity, making them promising scaffolds for future drug development. Moreover, their natural origin offers distinct advantages, including improved safety and biocompatibility.

The decline in recent research also reflects broader challenges, such as limited funding and a reduced focus on natural product-based drug discovery. This highlights the importance of reinvigorating interest in phytochemicals by leveraging modern analytical and computational tools. Renewed efforts in this direction could unlock new opportunities to develop these compounds as effective and sustainable alternatives to synthetic inhibitors for Cathepsin K-related conditions, such as osteoporosis and cancer.

Despite the promising activities of phytochemicals, their relatively high IC_50_ values compared to synthetic inhibitors highlight a significant limitation. For example, synthetic inhibitors such as Balicatib and Odanacatib achieve nanomolar IC_50_ values (Table 1), whereas most phytochemicals remain in the micromolar range (Table 3). This disparity underscores the need for structural optimization to enhance the binding affinity and specificity of phytochemicals. To overcome these challenges, strategies such as advanced computational modeling, structural modification through medicinal chemistry, and high-throughput screening of derivatives could be employed. Moreover, while most studies on phytochemicals have been limited to in vitro assays, future research should prioritize cell-based assays, animal models, and clinical trials to validate their therapeutic potential. By addressing these gaps, phytochemicals could transition from exploratory candidates to viable therapeutic agents targeting Cathepsin K-mediated diseases. This systematic approach would not only enhance the clinical applicability of phytochemicals but also contribute to the broader understanding of natural products as therapeutic resources for challenging biological targets.

**Table 3 molecules-30-00091-t003:** Phytochemicals as inhibitors of Cathepsin K: sources, structures, and bioactivities. This table summarizes the natural compounds identified as inhibitors of Cathepsin K, highlighting their sources, structural classes, and inhibitory activities.

Inhibitor	Source	Class	IC_50_	Details:
AC-5-1 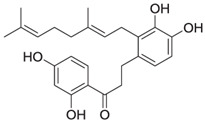	*Artocarpus altilis* (breadfruit)	Flavonoid	170 nM	Exhibits significant inhibitory activity against Cathepsin K, confirmed through in vitro studies [81].
Cycloaltilisin 6 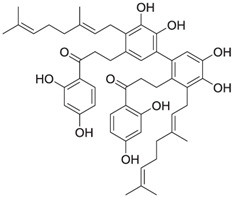	*Artocarpus altilis* (breadfruit)	Dimeric flavonoid	98 nM	Contains unique structural linkages contributing to its potent Cathepsin K inhibition [81].
Cycloaltilisin 7 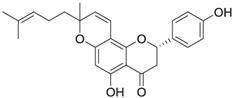	*Artocarpus altilis* (breadfruit)	Dimeric flavonoid	840 nM	Demonstrates moderate Cathepsin K inhibition; characterized by advanced NMR techniques [81].
Nicolaioidesin C 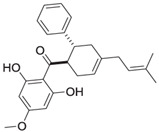	*Boesenbergia pandurata* (fingerroot)	Cyclohexenyl chalcone	4.4 µM	Selective Cathepsin K inhibitor, synthesized racemically for bioactivity studies [83].
Panduratin A 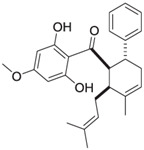	*Boesenbergia pandurata* (fingerroot)	Cyclohexenyl chalcone	5.1 µM for (+)-panduratin A 6.6 µM for (−)-panduratin A	Potent inhibitor of Cathepsins G, H, K, L, S, V, and X; highly cytotoxic against prostate cancer cells [83].
HIF 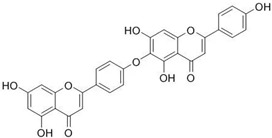	*Cycas guizhouensis*	Biflavone	1.54 µM	Reversible inhibitor of Cathepsin B and Cathepsin K. Exhibits selective inhibitory activity against Cathepsin B endopeptidase and forms hydrophobic interactions with CatB active sites [84].
AMF4 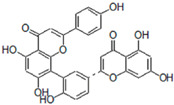	*Cycas guizhouensis*	Biflavone	1.39 µM	Reversible inhibitor of Cathepsin B and Cathepsin K [84].
6-Shogaol 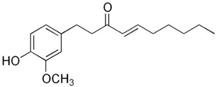	*Zingiber officinale* (ginger rhizome)	Phenolic compound	Ki = 16.6 µM (uncompetitive)	Exhibits uncompetitive inhibition of Cathepsin K, showing potential as an anti-inflammatory agent [86].

## 6. Insights into Cathepsin K Inhibition by Phytochemicals: Docking Studies and Optimization Strategies

Table 3 summarizes eight types of phytochemicals identified as inhibitors of Cathepsin K. However, docking studies have not yet been conducted for most of these compounds. Docking studies are a powerful computational tool that simulates the interactions between small molecules and their target proteins, offering insights into binding mechanisms, interaction sites, and potential inhibitory efficacy [87,88]. These studies provide critical information for optimizing the design of more potent inhibitors and identifying structural features essential for binding specificity and efficacy.

As shown in Figure 4, the docking results of Cathepsin K inhibitors AC-5-1, Cycloaltilisin 6, and Cycloaltilisin 7, analyzed using CB-Dock2 [89,90], provide detailed insights into their binding mechanisms within the Cathepsin K active site. Among the three, Cycloaltilisin 6 demonstrated the strongest binding affinity with a docking score of −8.5 and an IC_50_ value of 98 nM. This potent inhibition can be attributed to its dimeric flavonoid structure, which enables extensive interactions with critical active site residues, including Cys25, Asn161, and His162. Cycloaltilisin 6 forms multiple stabilizing hydrogen bonds, van der Waals interactions, and pi–alkyl interactions, contributing to its high specificity and strong inhibitory effect.

Cycloaltilisin 7, with a docking score of −6.9 and an IC_50_ value of 840 nM, exhibited a better binding affinity than AC-5-1, despite its higher IC_50_ value. Cycloaltilisin 7 shares a dimeric flavonoid structure similar to Cycloaltilisin 6, allowing for interactions with key residues such as Cys25 and His162. However, fewer stabilizing interactions, such as hydrogen bonds and hydrophobic contacts, result in lower inhibitory potency compared to Cycloaltilisin 6.

AC-5-1 displayed the weakest docking score among the three at −6.2 and an IC_50_ value of 170 nM. The compound interacts with essential active site residues such as Cys25 and His162 but has a simpler flavonoid structure, limiting the number and strength of its stabilizing interactions. Although its IC_50_ value is lower than Cycloaltilisin 7, the reduced binding affinity observed in docking studies underscores the importance of structural complexity in effective inhibition.

This analysis underscores the critical role of molecular interactions within the Cathepsin K active site in determining inhibitory efficacy. Cycloaltilisin 6 stands out for its strong binding affinity and structural complexity, while Cycloaltilisin 7 demonstrates moderate potential with a slightly better docking score than AC-5-1. However, as indicated by comparisons with synthetic inhibitors in Table 1, these natural product-derived inhibitors generally exhibit higher IC_50_ values. This gap highlights the necessity for structural optimization of phytochemicals. Guided by docking studies and structure–activity relationship (SAR) analyses, such modifications could enhance binding affinity, lower IC_50_ values, and improve specificity toward the Cathepsin K active site. By addressing these challenges, optimized phytochemicals could transition from promising in vitro candidates to clinically viable therapeutic agents for diseases associated with Cathepsin K activity.

## 7. Conclusions

Phytochemicals offer a promising avenue for developing Cathepsin K inhibitors due to their natural origin, structural diversity, and favorable safety profiles. Compounds such as Cycloaltilisin 6 and Panduratin A demonstrate strong binding affinities and moderate IC_50_ values, supporting their potential as therapeutic agents. However, compared to synthetic inhibitors, these phytochemicals face limitations, including lower potency and a lack of extensive in vivo and clinical studies. Future research must focus on enhancing their binding affinities through structural modifications, employing advanced computational modeling, and validating efficacy in animal models and clinical trials. By addressing these challenges, phytochemical-based inhibitors could transition from exploratory candidates to viable therapeutics for treating bone disorders like osteoporosis, offering a safer alternative to synthetic drugs.

## Figures and Tables

**Figure 1 molecules-30-00091-f001:**
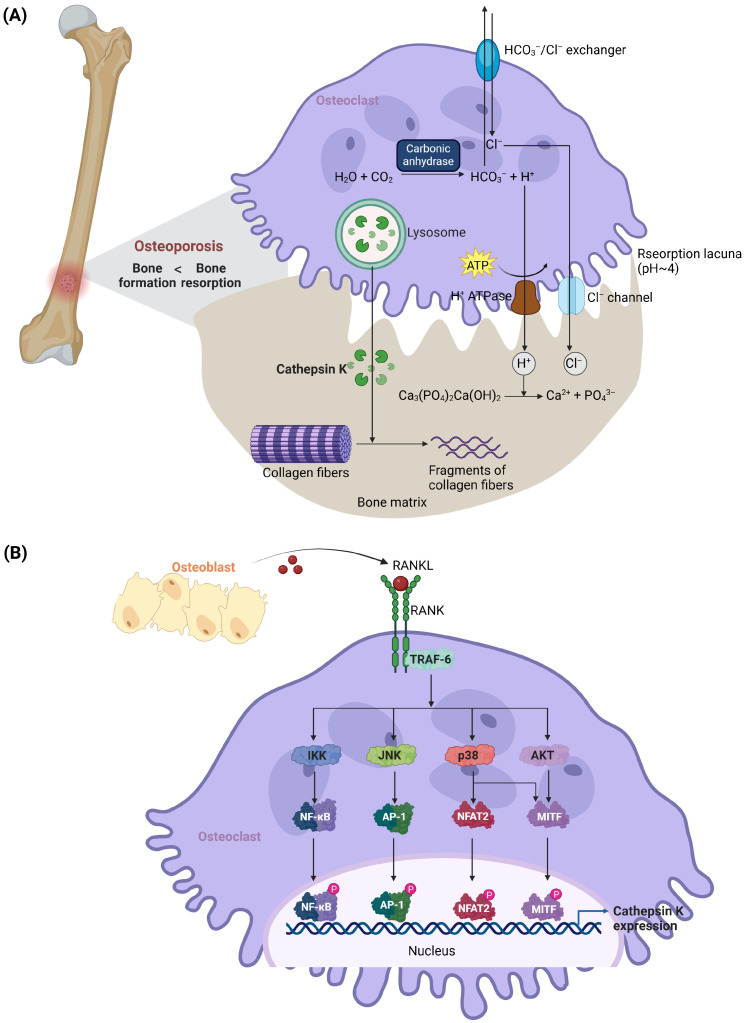
Mechanisms of bone resorption and regulation of Cathepsin K expression in osteoclasts. (**A**) Bone resorption by osteoclasts occurs within a specialized microenvironment called the resorption lacuna. The acidic pH (~4) in this compartment is established by the active transport of hydrogen ions (H^+^) via H^+^-ATPase, while chloride ions (Cl^−^) maintain electrochemical balance. This acidic environment dissolves hydroxyapatite, releasing calcium (Ca^2+^) and phosphate (PO_4_^3−^) ions. Concurrently, Cathepsin K, secreted by osteoclasts, degrades Type I collagen, the primary component of the bone matrix. Degraded collagen fragments are absorbed and processed by osteoclasts, enabling normal bone remodeling. (**B**) The regulation of Cathepsin K expression is mediated by the RANKL–RANK-signaling pathway. RANKL, secreted by osteoblasts, binds to its receptor RANK on osteoclast precursors, activating downstream pathways such as NF-κB, MAPK, and MITF. These pathways synergistically enhance the transcription of Cathepsin K, supporting osteoclast-mediated bone resorption. Dysregulation of these mechanisms is associated with pathological bone loss in conditions like osteoporosis.

**Figure 2 molecules-30-00091-f002:**
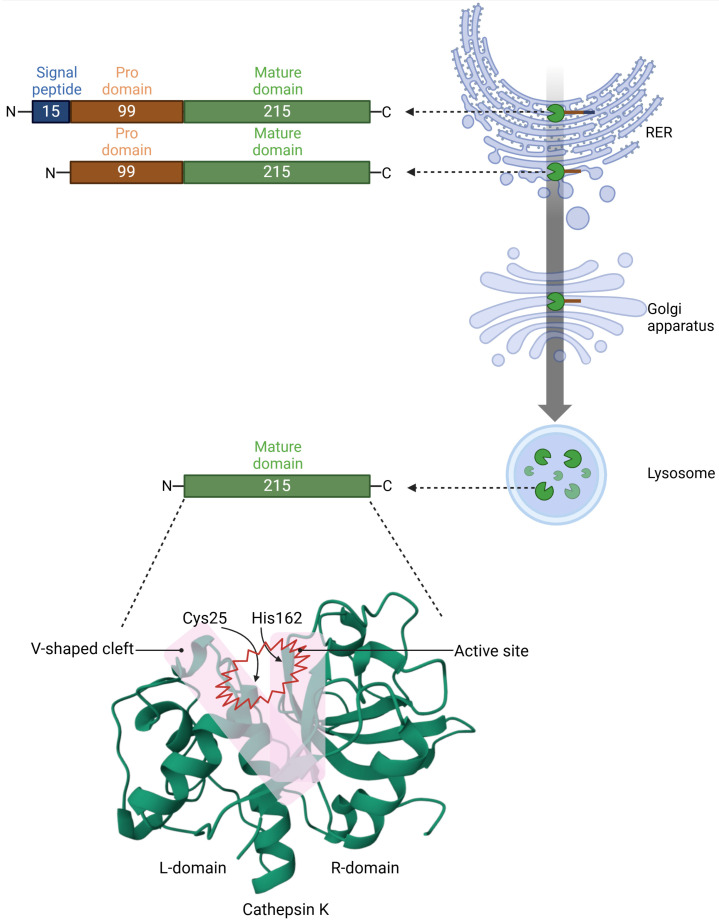
Biosynthesis, processing, and active site structure of Cathepsin K. The biosynthesis, post-translational processing, and structural features of Cathepsin K are shown. The pre-proenzyme form of Cathepsin K consists of 329 amino acids, including a 15-amino acid signal peptide, a 99-amino acid propeptide domain, and the 215-amino acid mature catalytic domain. The signal peptide directs the nascent protein to the rough endoplasmic reticulum (RER), where it is cleaved. The proenzyme is subsequently transported to the Golgi apparatus for further processing and targeted to lysosomes via the mannose-6-phosphate-receptor pathway. Within the lysosome, the propeptide is cleaved between Arg114 and Ala115, resulting in the mature, active enzyme. The mature domain of Cathepsin K is structurally organized into two domains: the L-domain, characterized by a central helix, and the R-domain, dominated by beta-barrel motifs. Together, these domains form a V-shaped active site cleft. The catalytic residues, Cys25 and His162, located at the interface of the two domains, facilitate the enzymatic cleavage of collagen and other substrates, underscoring the enzyme’s role in bone resorption and extracellular matrix remodeling. The 3D structure of Cathepsin K shown below was obtained from the Protein Data Bank (https://www.rcsb.org/ (accessed on 8 October 2024), with the ID: 1ATK.

**Figure 3 molecules-30-00091-f003:**
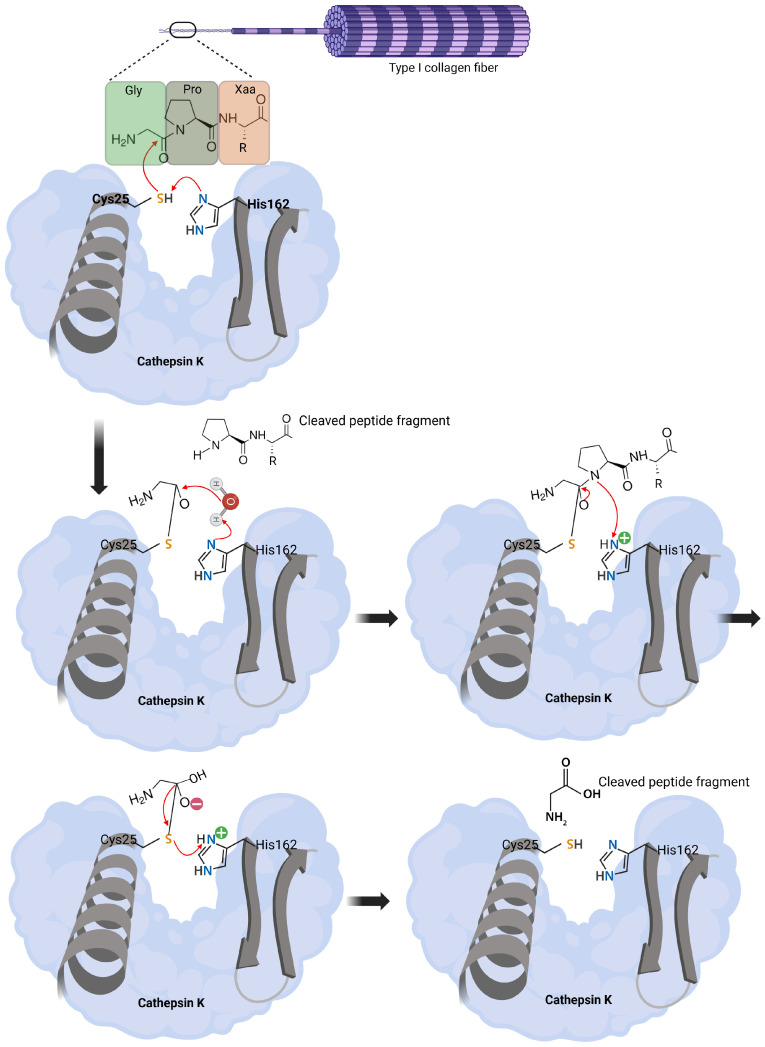
Catalytic mechanism of Cathepsin K in collagen degradation. The catalytic mechanism of Cathepsin K, a cysteine protease, during the degradation of Type I collagen fibers is depicted. The process begins with the nucleophilic attack of the thiolate group of the catalytic residue Cys25 on the carbonyl carbon of the peptide bond within the Gly–Pro–Xaa sequence of collagen. This attack is facilitated by the imidazole group of His162, which acts as a general base by abstracting a proton from the thiol group of Cys25, thereby activating it. The nucleophilic attack leads to the formation of a tetrahedral intermediate, where the carbonyl oxygen acquires a negative charge that is stabilized by the active site environment. Following the formation of the tetrahedral intermediate, the peptide bond between Gly and Pro is cleaved, resulting in the release of the peptide fragment. The catalytic cycle continues with the entry of a water molecule into the active site, which is also activated by His162. The water molecule attacks the acyl–enzyme intermediate, hydrolyzing it and regenerating the thiolate group of Cys25. This reaction leads to the release of the C-terminal peptide fragment and restores the enzyme to its original state, ready for another catalytic cycle. The active site residues Cys25 and His162 are essential for the catalytic activity of Cathepsin K, ensuring efficient cleavage of the collagen substrate. This mechanism highlights the unique ability of Cathepsin K to degrade the triple-helical structure of collagen by targeting specific sequences, thereby playing a critical role in bone resorption and extracellular matrix remodeling. The cleaved peptide fragments are indicated to emphasize the products of the catalytic process.

**Figure 4 molecules-30-00091-f004:**
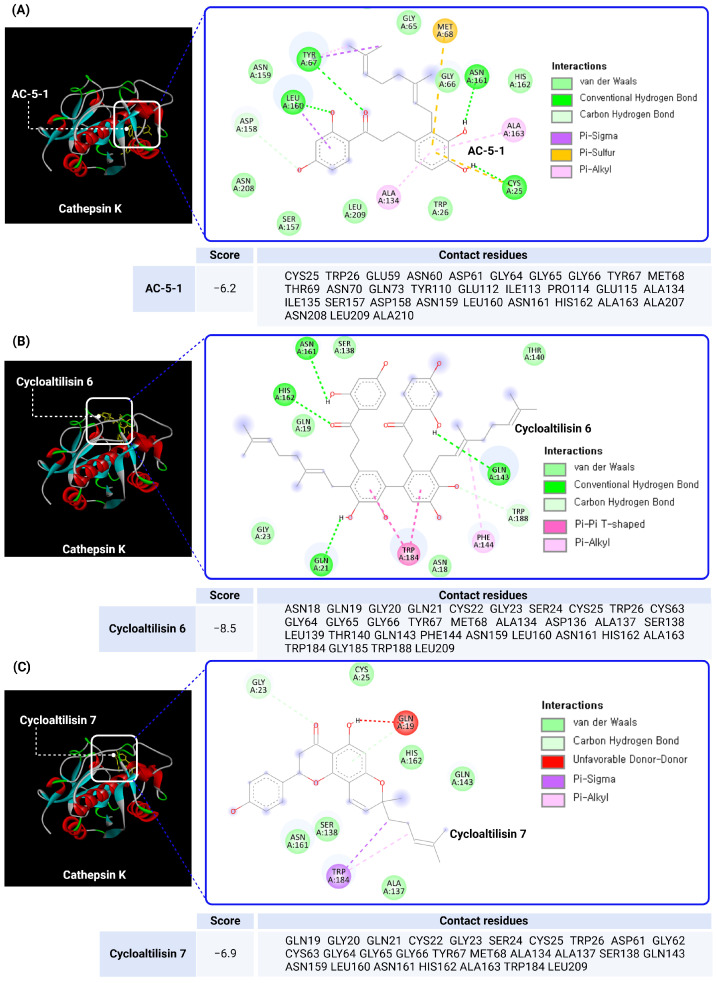
Docking analysis of phytochemical inhibitors (AC-5-1, Cycloaltilisin 6, Cycloaltilisin 7) binding to the active site of Cathepsin K. The docking interactions of three phytochemical inhibitors, (**A**) AC-5-1, (**B**) Cycloaltilisin 6, and (**C**) Cycloaltilisin 7, within the active site of Cathepsin K are illustrated. The protein structure of Cathepsin K was retrieved from the Protein Data Bank (PDB ID: 1ATK), and docking simulations were performed using CB-Dock2 (https://cadd.labshare.cn/cb-dock2/index.php (accessed on 15 October 2024). The chemical structures of the inhibitors were sourced from PubChem (https://pubchem.ncbi.nlm.nih.gov/ (accessed on 15 October 2024), ensuring accurate molecular configurations for docking analysis. Key binding interactions, including hydrogen bonds, van der Waals forces, Pi–alkyl interactions, and unfavorable donor–donor interactions, are highlighted alongside their associated contact residues. Cycloaltilisin 6 demonstrated the strongest binding affinity with a docking score of −8.5, supported by extensive interactions with critical residues such as Cys25, HisS162, and Asn161. AC-5-1 displayed a moderate binding affinity with a docking score of −6.2, forming stabilizing interactions with residues like Cys25 and His162.

## Data Availability

The data presented in this study are available on request from the author.

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
