# Peer review of "Review of Cathepsin K Inhibitor Development and the Potential Role of Phytochemicals"

_molecules, 2024, doi:10.3390/molecules30010091_

Round 1
Reviewer 1 Report
Comments and Suggestions for Authors
The authors present a comprehensive review about cathepsin K inhibitors development and the potential of phytochemicals to be used as inhibitors of this enzyme. However, there are some points to be considered.
1.- The figure legends have to be improved, in its actual form are too long and some of the information described is included in the main text.
2.- In table 3 or as separate image, please include the structure of all phytochemicals mentioned.
3.- Phytochemicals section must to be improved, please include more information about their inhibition, not just their IC50 value, in table 3 additional information is included in the las column. These data can be described in more detail in the main text and delet thsi column fro the table.
4.- References in phytochemicals section are limited and between 10 or 20 years old. These are the only work available? If this is true, please include in the main text a potential explanation about it and how this affect to encourage the use of phytichemicals as future drug candidates against cathepsin K.
Author Response
1.- The figure legends have to be improved, in its actual form are too long and some of the information described is included in the main text.
⟶ I have revised the legend for Figure 1 according to your request.
2.- In table 3 or as separate image, please include the structure of all phytochemicals mentioned.
⟶ I have addressed your request by including the chemical structures of all phytochemicals mentioned in Table 3.
3.- Phytochemicals section must to be improved, please include more information about their inhibition, not just their IC50 value, in table 3 additional information is included in the las column. These data can be described in more detail in the main text and delet thsi column fro the table.
⟶ I have enhanced the Phytochemicals section by providing more detailed information.
4.- References in phytochemicals section are limited and between 10 or 20 years old. These are the only work available? If this is true, please include in the main text a potential explanation about it and how this affect to encourage the use of phytichemicals as future drug candidates against cathepsin K.
⟶ The references in the Phytochemicals section primarily span the early 2000s to early 2010s due to the limited availability of recent studies, and I have included an explanation in the main text addressing this and its implications for encouraging the exploration of phytochemicals as future drug candidates against cathepsin K.
Reviewer 2 Report
Comments and Suggestions for Authors
Dong Oh Moon highlights that phytochemicals represent a promising alternative for the development of cathepsin K inhibitors, offering advantages such as their natural origin, structural diversity, and favorable safety profiles. Key compounds, such as Cycloaltilisin 6 and Panduratin A, demonstrate significant binding affinities and moderate inhibitory potencies, suggesting their potential as therapeutic agents for bone-related diseases. However, these phytochemicals currently face challenges, including lower potency compared to synthetic inhibitors and limited in vivo and clinical data. To advance their clinical applications, future research should focus on optimizing their binding affinities through structural modifications, utilizing advanced computational modeling, and conducting comprehensive in vivo and clinical trials. By overcoming these challenges, phytochemical-based inhibitors could provide a safer, more effective alternative to synthetic drugs for managing cathepsin K-related diseases, such as osteoporosis. This manuscript meets the submission standards for the journal, pending minor revisions.
Comment:
- Rewrite all references in the manuscript according to the Molecules journal format for clarity and consistency.
Author Response
- Rewrite all references in the manuscript according to the Molecules journal format for clarity and consistency.
⟶ I have reformatted all references in the manuscript to align with the Molecules journal guidelines for clarity and consistency.
Reviewer 3 Report
Comments and Suggestions for Authors
The article reviews Cathepsin K inhibitors. The article rationale is adequate; the literature review is almost complete. The figures are informative, and the new development is short but within the line of expectations. The only suggestions I have are: 1) most of the compounds of Table 1 inhibit other cathepsins, the IC50 should be added to the Table, and 2) Odanacatib was a promising compound; however, the incidence of stroke was higher in the treated individuals, this information should be stated in the Table. The same point should be stated for the rest of the compounds on clinical trials and those new compounds being developed.
Author Response
The article reviews Cathepsin K inhibitors. The article rationale is adequate; the literature review is almost complete. The figures are informative, and the new development is short but within the line of expectations. The only suggestions I have are: 1) most of the compounds of Table 1 inhibit other cathepsins, the IC50 should be added to the Table, and 2) Odanacatib was a promising compound; however, the incidence of stroke was higher in the treated individuals, this information should be stated in the Table2. The same point should be stated for the rest of the compounds on clinical trials and those new compounds being developed.
⟶ I have updated Table 1 to include IC50 values for other cathepsins and revised Table 2 to include information about the increased risk of stroke associated with Odanacatib, as requested.